# One-Step Solvothermal Synthesis by Ethylene Glycol to Produce N-rGO for Supercapacitor Applications

**DOI:** 10.3390/nano13040666

**Published:** 2023-02-08

**Authors:** Mohammad Obaidur Rahman, Nursyarizal Bin Mohd Nor, Narinderjit Singh Sawaran Singh, Surajudeen Sikiru, John Ojur Dennis, Muhammad Fadhlullah bin Abd. Shukur, Muhammad Junaid, Ghulam E. Mustafa Abro, Muhammad Aadil Siddiqui, Md Al-Amin

**Affiliations:** 1Department of Electrical & Electronic Engineering, Universiti Teknologi PETRONAS, Seri Iskandar 32610, Perak, Malaysia; 2Faculty of Data Science and Information Technology (FDSIT), INTI International University, Persiaran Perdana BBN, Putra Nilai, Nilai 71800, Negeri Sembilan, Malaysia; 3Centre for Subsurface Imaging, Universiti Teknologi PETRONAS, Seri Iskandar 32610, Perak, Malaysia; 4Department of Fundamental & Applied Science, Universiti Teknologi PETRONAS, Seri Iskandar 32610, Perak, Malaysia; 5Centre of Innovative Nanostructure and Nanodevices (COINN), Universiti Teknologi PETRONAS, Seri Iskandar 32610, Perak, Malaysia; 6Department of Electronic Engineering, Balochistan University of Information Technology, Engineering and Management Sciences, Quetta 87300, Balochistan, Pakistan; 7The University of Queensland, St Lucia, QLD 4072, Australia

**Keywords:** supercapacitor, nitrogen doping, electrode materials, solvothermal, reduced graphene oxide, organic solvent

## Abstract

Graphene and its derivatives have emerged as peerless electrode materials for energy storage applications due to their exclusive electroactive properties such as high chemical stability, wettability, high electrical conductivity, and high specific surface area. However, electrodes from graphene-based composites are still facing some substantial challenges to meet current energy demands. Here, we applied one-pot facile solvothermal synthesis to produce nitrogen-doped reduced graphene oxide (N-rGO) nanoparticles using an organic solvent, ethylene glycol (EG), and introduced its application in supercapacitors. Electrochemical analysis was conducted to assess the performance using a multi-channel electrochemical workstation. The N-rGO-based electrode demonstrates the highest specific capacitance of 420 F g^−1^ at 1 A g^−1^ current density in 3 M KOH electrolyte with the value of energy (28.60 Whkg^−1^) and power (460 Wkg^−1^) densities. Furthermore, a high capacitance retention of 98.5% after 3000 charge/discharge cycles was recorded at 10 A g^−1^. This one-pot facile solvothermal synthetic process is expected to be an efficient technique to design electrodes rationally for next-generation supercapacitors.

## 1. Introduction

Ultracapacitors (UCs) or electrochemical capacitors, widely known as supercapacitors (SCs), are considered as next-generation energy storage technology owing to some of their unique properties such as high power density, fast charging/discharging efficacy, long cycle stability (>100,000 cycles), high rate capability, low maintenance cost, product safety, and environmental friendliness. The applications of supercapacitors are increasing day by day with the rapid expansion of the world market for portable electronics, electrical automobiles, and stationary energy storage systems [1,2,3,4,5,6,7]. Typically, a physical supercapacitor cell is composed of two electrodes immersed in an electrolyte separated by a separator. For symmetric cells, the electrodes are identical, whereas for asymmetric cells, they can be distinct. The separator soaked in an electrolyte prevents electrical contact between the electrodes. To obtain the optimal performance, the separator materials should be ion-permeable, allowing the transfer of ionic charge with a high ionic conductance while also having a high electrical resistance and small thickness [8,9]. Based on the charge storage phenomenon and cell configuration in SC devices, electric double-layer capacitors (EDLCs), pseudocapacitors (PCs), and hybrid capacitors can be characterized [10,11]. The working principle of EDLCs is to store charges at the electrode–electrolyte interface with a high power density where there is no transfer of electrons into species (i.e., adsorption–desorption occurs by electrochemical charge accumulation via a non-Faradaic process) owing to lower internal resistance. PCs exploit a fast and reversible oxidation and reduction process between the electroactive species in electrodes and electrolytes (i.e., surface redox reaction occurs across the electrode–electrolyte interface) [12,13]. The nano-porous carbon materials with a high specific surface area, especially for EDLCs and metal oxides, as well as conducting polymers, sometimes functionalized porous carbon for PCs, are used to prepare the supercapacitor electrodes [14]. The specific capacitance of EDLCs is strongly dependent on the surface area of electrode materials that are accessible to the electrolyte. In the case of PCs, they mostly undergo the electrochemical Faradaic process [15,16]. Hence, a rational electrode design with sufficient electric conductivity and a hierarchical charge-transfer structure is crucial to achieving a high specific capacitance and energy density for supercapacitors, but still, it remains a challenge [17,18].

Graphene, a single-atom-thick, 2D carbon allotrope bonded in a hexagonal honeycomb lattice of sp^2^ carbons, has been demonstrated as a promising electrode material for supercapacitor electrodes owing to the combination of various distinct features such as large surface area (2630 m^2^ g^−1^) [19], high thermal conductivity (5000 W m^–1^ K^−1^), suitable electron mobility (250,000 cm^2^ V^−1^ s^−1^) at room temperature [20], high mechanical strength (42 N/m breaking strength), high electrochemical stability, and adequate electrical conductivity (10^4^ S/cm). Several methods are used to produce graphene, including the reduction of graphene oxide (rGO), micromechanical exfoliation of graphite, chemical vapor deposition (CVD), epitaxial growth, chemical intercalation, and chemical vapor deposition (CVD). The preparation of graphene via GO reduction stands out among these techniques because it shows potential for the bulk manufacture of products based on graphene [21]. For the mass production of GO, a chemical synthesis process can be employed to oxidize the inexpensive graphite flakes, assisted by strong acids and oxidants [22], resulting in GO containing numerous oxygen-related functional groups on the basal plane or edge layer. It exhibits low conductivity because of the broken π–π conjugation in the graphite composition. The broken graphitic layers can be repaired using a reducing agent and thus, high conductivity is possible to achieve.

To employ graphene in potential applications such as supercapacitors, it is essential to regulate its electronic properties carefully [15]. The performance of graphene-based electrodes can be affected by the change in molecular structure, such as (1) the addition of pseudocapacitive moieties such as carbonyl or hydroxyl groups and (2) doping heteroatoms (N, B) to increase the quantum capacitance in graphene morphology, which results in the state density of electrons of graphene being changed, their two-dimensional planner network being distorted, and a barrier within the electron-potential continuum being created. The performance of graphene supercapacitors could be optimized if a compromise between conductive conjugate channels and heteroatom introduction is made. In this context, Fang et al. improved an acid-assisted ultrafast thermal-processing mechanism to enhance the oxygen functional group content of graphene. These functional groups not only developed the electrolyte’s wettability to the graphene surface and facilitated the structure of the electrical double layer, but also presented additional pseudo-capacitance [23,24]. Recently, numerous studies showed that substituting carbon atoms in graphene with nitrogen can efficiently tune its intrinsic properties, including surface area, chemical stability, and conductivity, hence increasing the performance [25]. Additionally, while doping graphene with nitrogen, the electronic band structure of graphene is exchanged with its extra valence electrons by creating novel energy bands in the lower portion of the conduction band of sp^2^-bonded carbon atoms [26]. Moreover, the N-atom contains three valence electrons to facilitate powerful bonding, and the extra pair of electrons enhances the total conductivity [27]. Due to the similarities of nitrogen’s atomic structure and powerful valence bonds to the carbon species, it became a significant element for the chemical doping of graphene-based materials [28].

Usually, graphene is doped with heteroatoms by direct synthesis procedures, including the arch-discharge method, solvothermal process, segregation growth, chemical vapor deposition (CVD) process, and hydrothermal method. Some are post-treatment synthesis procedures such as plasma treatment process, hydrazine hydrate, and thermal and chemical treatment [15,29]. Some of these methods demand high-standard equipment that is highly expensive, and during bulk production, several difficulties can be introduced. Solvothermal reaction assisted by an appropriate solvent and a reducing agent is a useful way to overcome the above-mentioned issues because of their cost-effectiveness and facile synthetic process [22]. Ethylene glycol (EG) has some desirable features as a solvent such as adequate solubility, low cost, relatively high boiling point, low toxicity, and certain reducibility [30]. Recently, ammonia and urea have been extensively used as the nitrogen source in graphene doping and hydrazine hydrate as a reducing agent [31,32].

In this work, we produced N-doped reduced graphene oxides (N-rGOs) with a scalable and facile solvothermal synthesis procedure assisted by an odorless and colorless organic solvent, ethylene glycol (EG), and a powerful reducing agent, hydrazine hydrate, which effectively reduced the oxygen-related functional group in GO at the reaction temperature of 180 °C for 12 h, as graphically illustrated in Figure 1. The use of this organic solvent in our experiment is more suitable than other harmful and hazardous chemicals, avoiding complex material handling with special types of instruments. The electrochemical characterization of the N-rGO electrode was examined by the Autolab PGSTAT302N electrochemical workstation connected with a three-electrode cell in 3 M KOH electrolyte. Due to the significant number of nitrogen atoms injected into the carbon lattice and the highly reduced oxygen-related functional groups from the GO, the N-rGO materials displayed suitable specific capacitance (420 F g^−1^ at 1 A g^−1^) and better cyclability with adequate rate capability. As far as we know, this report is novel, revealing the synthesis process of nitrogen-rich graphene nanoparticles using the organic solvent EG.

## 2. Experimental Section

Urea (CH_4_N_2_O, 99.0–100.5%), hydrazine hydrate solution (N_2_H_4_.H_2_O 80%), and ethylene glycol (C_2_H_6_O_2_ 99.8%) were analytical-grade reagents purchased from Merck, Kenilworth, NJ, USA, and applied for further experimentation.

### 2.1. G.O. Synthesis

Natural graphite flakes were successfully oxidized to produce GO through an improved Hummer method reported in our previous work [33,34]. In brief, in 100 mL of sulfuric acid (H_2_SO_4_), 4 g of K_2_FeO4, 6 g of KMnO_4_, 10 g of flake graphite, and 0.01 g of boric acid were mixed into a round bottom flask and stirred for 2 h, maintaining the temperature at 5 °C. After that, the flask was placed into an ice bath at around 35 °C to avoid explosion from overheating. An additional 5 g of KMnO_4_ was added and stirred for another 10 h to maximize the oxidation and exfoliate the graphite into several layers. Subsequently, the temperature was maintained at about 95 °C, and 250 mL of deionized (DI) water was mixed slowly; we waited for 15 min until the suspension color changed to brown. Again, this brown color precipitate was treated with 12 mL of H_2_O_2_ to remove the residual oxidants. The product was further washed with HCl (10%), followed by the dilution of the solution in DI water and centrifugation at 6000 rpm to collect GO. The flake GO was dried in a freeze dryer.

### 2.2. Synthesis of N-rGOs

N-rGO-1 was prepared via a one-step solvothermal process applying EG as an organic solvent. To make a GO-based suspension, 75 mg of GO was mixed in 50 mL of DI water and ultrasonicated for 45 m. Subsequently, 2 g of urea, 6 mL of N_2_H_4_.H_2_O, and 50 mL of EG solution were mixed into the above mixer and magnetically stirred for another 4 h. Then, the mixture was taken to a 200 mL stainless steel Teflon-lined autoclave and heat-treated for 12 h at the set temperature of 180 °C, as shown in Figure 1. The as-prepared black precipitate of N-rGO1 was washed and filtered several times by DI water and ethanol to remove unreacted substances and impurities after naturally cooling down to room temperature. Finally, the product was freeze-dried for 24 h and collected for further use. The N-rGO-2 was prepared according to the same process with different mixing quantities of material, i.e., 50 mg of GO, 3 g of urea, and 10 mL of hydrazine hydrate solution.

### 2.3. Electrodes Fabrication and Performance Measurement

The electrodes for the supercapacitors were fabricated by depositing the N-rGO-based slurry on nickel (Ni) foam. The composite of the slurry was made by adding 80 wt % of as-prepared N-rGOs, 12 wt % of PVDF, and 8 wt % of super p (carbon black), deliquesced in a certain amount of NMP. The mixture was ultrasonicated for 30 min, followed by stirring for another 12 h, maintaining the temperature at 85 °C to produce a concentrated uniform blend. Ni foam substrates (3 × 1 cm) were cut into pieces and cleaned with HCl, ethanol, and DI water to use as the current collector. The slurry was painted onto it (area of 1 × 1 cm). The painted Ni foam was dried with a hot plate at 65 °C for 18 h. The measurements of cyclic voltammetry (CV), galvanostatic charge–discharge (GCD), and electrochemical impedance spectroscopy (EIS) were run by a PGSTAT302N (Metrohm auto lab) electrochemical workstation connected with a three-electrode cell in 3 M KOH electrolyte. The N-rGO, Pt mesh, and Ag/AgCl acted as working, counter, and reference electrodes, respectively. The details of the optimization and performance tests are displayed in Figure 2, and the specific capacitance ($C_{sp}$), energy density, and power density were determined using the equations below [35,36,37].
(1)$$C_{sp}=\frac{Q}{V}=\frac{\;\int_{V_{1}}^{V_{2}}I\left(V\right)dV}{mv\left(V_{2}-V_{1}\right)}$$
(2)$$C_{sp}=\frac{I\times \Delta t}{\Delta V\times m}$$
(3)$$E=\frac{1}{2}CV^{2}$$
(4)$$P=\frac{E}{\Delta t}$$

In this equation, m is the mass of the materials, I is charge–discharge current, Δt is full discharge time, ΔV is full discharge potential change, E and P are specific energy and power, respectively.

## 3. Results and Discussion

### 3.1. Structural Features

Figure 3 depicts the FE-SEM images of GO, N-rGO-1, and N-rGO-2 and the elemental mapping of carbon and nitrogen for N-rGOs. The FE-SEM images of GO and N-rGOs revealed that the well-separated platelets tightly bound together.

Figure 3a,d shows the exfoliated layers of GO, indicating the existence of carbon and oxygen. We further doped previously prepared GO with N molecules by the solvothermal synthetic process. Several wrinkles and corrugations were realized, indicating the oxygen-containing functional group and dopant elements (N). In Figure 3c, the apparent wrinkles and pigmented surface on N-rGO-layers confirm the footprints of N-doping on the surface morphology of N-rGOs. Furthermore, Figure 3e,f shows the elemental mapping of nitrogen and carbon of N-rGOs in which a uniform dispersion of material elements in the N-rGO-samples is indicated. This finding supports the effective synthesis of N-rGOs.

### 3.2. XRD

Figure 4 depicts XRD patterns of GO and N-rGOs. The exfoliated GO processed from the chemical oxidation of graphite flakes shows a stronger diffraction peak at 2θ = 11.0° associated with the 001 plane and has inter-layer basal spacing of 0.80 nm, which is higher than that of intrinsic graphite (0.34 nm). Different oxygen-containing functional groups are introduced into the basal space and onto the edge of the graphite structure during oxidation. Carbonyl, hydroxyl, and epoxy groups are added to the graphene sheets’ basal plane, while carboxyl groups are added to their edges. In contrast to intrinsic graphitic structures, hydrophilic water can be intercalated inside the stacked GO morphology, resulting in a wide basal gap. The removal of the oxygen-related functional groups of the water molecules’ carbon layers of the GO results in the creation of monolayer graphene and a decrease in d-spacing [38].

After the solvothermal treatment, small and broad peaks centered at 2θ = 24.3° and 2θ = 43.0° for N-rGOs were realized, corresponding to the 002 and 001 planes, respectively, instead of the sharp peak centered at 2θ = 10.5° for GO, indicating that the GO had been exfoliated and restacked during the doping process due to van der Waals and electrostatic forces. A shift in the peaks results in a bigger angle; as a result, the hexagonal structure’s unit cell volume decreases. In addition to the observed peak shift, the peaks become wider and vanish as the reduction temperature increases.

### 3.3. FTIR Spectroscopy

Spectroscopy is an analytical method used to investigate the oxygen-containing functional groups of GO and N-rGO at various thermal reduction temperatures. There are several oxygen configurations in the structure, including the vibrational modes of epoxide (C-O-C) (1230–1320 cm^−1^) and C=O (1595–1650 cm^−1^), and carboxyl (COOH) (1650–1750 cm^−1^), carbonyl (C=O) (1700–1800 cm^−1^), and hydroxyl (C-OH) (3050–3800 cm^−1^) groups. The peaks show the combined contribution of the C-O stretching and O-H deformation vibrations at 1225 cm^−1^ (vigorous intensity). Moreover, symmetric C-H stretches (CH3) at 2853 cm^−1^, asymmetric C-H stretches (CH2) at 2925 cm^−1^, and asymmetric C-N stretches at 2960 cm^−1^ were found. These findings support the presence of several oxygen functional groups before heat reduction (carbonyl, carboxyl, hydroxyl, and epoxide) in the FTIR spectra of the GO, as previously discovered by several investigations. At 100 °C, the peak intensity related to C-O stretching (840 cm^−1^) disappears overnight [39]. The strength of the peak associated with carboxyl (COOH) stretching vibration in 1720 cm^−1^ has decreased. The strength of the peak at 1225 cm^−1^ is related to the combination of O-H deformation and C-O stretching vibrations; however, it disappears after the thermal reduction at 150 °C. Notably, the carbonyl (C=O) peak in GO at 1625 cm^−1^ shifts to a lower wave number (1565 cm^−1^) for rGO-120 owing to GO de-oxygenation and then remains constant after heat treatment [40]. Following oxidation and solvothermal synthesis, the GO and N-rGO were analyzed using Fourier transform infrared (FTIR) spectroscopy to recognize the functional groups in the samples (Figure 5). A large peak (between 3000 and 3700 cm^−1^) is seen in the GO spectra as a marker of the hydroxyl group and surface-adsorbed water [41,42].

Moreover, the numerous peaks at 1715, 1415, and 1227 cm^−1^ were designated as ketone (C=O), carboxyl (COOH), and epoxy (C-O-C) groups, respectively [43,44]. In N-rGOs spectra, a pair of new peaks existed in the 1000–1700 cm^−1^ range, indicating the C-N co-valent bonding in N-rGOs. Especially in the synthesis procedure, the wide spectrum of the -OH group was reduced to a profound range, and the peaks associated with N-C at 1605 and 1265 cm^−1^ were realized [45].

### 3.4. XPS

The chemical compositions of the functional groups of GO and N-rGOs were analyzed by XPS (Figure 6). The XPS data of GO were also documented for comparison purposes. The peaks in the N-rGOs survey spectrum at 285.4, 400.2, and 533.4 indicate C1s, N1s, and O1s spectra, respectively.

At the same time, in GO, only carbon (C-C) and oxygen (hydroxyl, epoxy, and carboxyl) species are visible (Figure 6a). A substantial decrease in O1s and an increase in C1s occurred. N1s in the N-rGO spectrum compared with GO corresponds to the reduction in oxide and hydroxide groups and doping with the foreign atom (N) in N-rGO samples [46,47]. The consequences show that a suitable content of nitrogen atoms (3.56%) is included in the graphene network (Table 1 details the content of each element). Figure 6b presents the convoluted C1s spectra of N-rGOs in which several peaks are visible, corresponding to C=C/C–C bonds (284.1 eV) of the sp^2^ carbon structures, C–N bonds (285.4 eV) of the carbonyl and epoxy groups, and C–O bonds (286.6 eV) of the carbonyl group associated with atomic bonding for carbon–carbon and carbon–oxygen connected functional groups. The carbon sp^2^ bonding in the 2D honeycomb lattice structure was attributed to the peak in C1s spectra observed at 284.1 eV [48,49].

Moreover, deconvoluted high-resolution N1s spectra of N-rGOs show four peaks at 398.1 eV (pyridinic-N), 400.3 eV (pyrrolic-N), 401.5 eV (graphitic-N), and 405.5 eV (oxidized-N) [50,51,52]. The pyridinic-N could significantly show a suitable pseudo-capacitance effect and enhance the conductivity, consequently increasing capacitive performance (Figure 6c) [53]. The deconvoluted O1s peaks from N-rGOs (Figure 6d) visible at 530.7, 531.9, and 533.4 eV are ascribed to the oxygen atoms with C–O bonds, C=O bonds, and O–C=O bonds, respectively [54].

### 3.5. Raman Spectroscopy

The Raman plot of GO, N-rGO-1, and N-rGO-2 samples is displayed in Figure 7. This method effectively determines the level of disorder in the materials [55]. The Raman spectra of GO and N-rGOs show two major Raman peaks at 1349 cm^−1^ and 1597 cm^−1^, which can be assigned to the D and G bands, respectively. The intensity peak found in the GO spectra is ascribed to the disorder and defects in the graphitic lattice due to the existence of oxygen-containing functional groups, and the two main Raman peaks in N-rGO samples indicate that graphene sheets had been doped with nitrogen. In N-rGO samples, the higher I_D_/I_G_ ratio (1.38 and 1.31) indicates the reduction in oxide groups and the presence of nitrogen due to the additional scattering effect caused by electron doping [56].

### 3.6. Electrochemical Performances

As mentioned in Section 2.3, the PGSTAT302N, Metrohm auto lab workstation was employed to determine the electrochemical performance of as-constructed N-rGO-based electrodes using a traditional three-electrode testing cell. During the test, cyclic voltammetry (CV), electrochemical impedance spectroscopy (EIS), and galvanostatic charge–discharge (GCD) techniques were used in 3 M KOH electrolyte. In the beginning, the CV analysis of the electrode from Nr-GOs was performed at numerous scan rates ranging from 2 to 100 mV s^−1^ in the potential range of −0.2 to + 0.5 V (0.7 V) (Figure 8a). In Table 2 the specific capacitance obtained from N-rGO sample is displayed with the comparisons of recently published other materials.

All curves obtained from the CV test demonstrated quasi-rectangular structures without any visible redox peak corresponding to the quasi-reversible properties. These findings strongly suggest that the quantum capacitance and its contribution to the interfacial capacitance are closely related to dopant concentration for monolayer graphene.

Another factor, the quantum capacitance, may, therefore, be relevant for some carbons having thin walls where the response of the charge inside the near-2D walls may have to be taken into consideration. Many researchers have adopted the pseudo-capacitance effect of the N functionalities from proton transfer as the only explanation for the change in interfacial capacitance for N-doped carbons when compared to the values of the pure C analogs [65].

We calculated the specific capacitance by Equation (1) from CV curves at different scan rates and obtained the highest capacitance of 500 F g^−1^ at a 2 mV s^−1^ scan rate with 3 M KOH electrolyte. We also noticed that the specific capacitances decreased with the increment in scan rates. The changes in capacitance with the change in scan rates are displayed in Figure 8c, which has a significant relationship with ion diffusion through the electrode. Ions in the electrolyte can have adequate time for diffusing the inner and outer surface of the electrodes at a low scan rate, with more charge accumulation increasing the capacitance. Elsewhere, the charge accumulation through the electrodes is highly affected, causing less capacitance due to the fast mobility of ions at high scan rates.

Figure 8b exhibits the curves of GCD of the N-rGO electrode recorded at the current densities of 1 to 6 A g^−1^ in the same voltage window mentioned above in CV measurements. The curves showed an almost symmetrical shape and adequate linearity in discharging slopes, indicating the quasi-reversible features. In addition, there was no significant IR drop, suggesting proper electrochemical reversibility and conductivity in the electrode materials. The highest acquired specific capacitance from the GCD curve for the N-rGO-based electrode was 420 F g^−1^ at 1 A g^−1^, which was calculated using Equation (2). Table 2 shows a comparison of the N-rGO-based electrode and other reported articles where N-rGOs have higher electrochemical performance. The EIS test was performed to determine the electrochemical accomplishment associated with the electrode–electrolyte interface at a frequency from 0.01 Hz to 100 kHz with a perturbation voltage of 10 mV. The conductivity and ion transit kinetics at the electrode–electrolyte interface are depicted in the Nyquist plot of N-rGO in 3 M KOH electrolyte in Figure 8d.

The EIS plot in Figure 8d comprises a semicircle in the higher-frequency spectrum, at 45° Warburg impedance in the medium-frequency spectrum, and a capacitive area across the low-frequency spectrum. The electrolyte’s resistance value (Rs) in the high-frequency band is close to 2.3 Ω, where the EIS line crosses the Z’ orbits. In the case of the low-frequency region, the EIS line has a smooth rise approaching 80° to the Z’ axis, corresponding to a suitable capacitive property. This can be described as ideal electrode–electrolyte interaction. Due to the charge transport resistance (Rct) from the interface of the electrolyte and electrode, the smaller semicircle in the high-frequency domain is visible. From the contact of ionic transport and the electrode–electrolyte interface, Rct begins. The realized lower Rct in the EIS data suggests the easy access of the ions in the electrolyte to the prepared N-rGO materials and the outstanding performance of the N-rGO electrode. Three thousand (3000) galvanostatic charge/discharge cycles at a high current density, i.e., 10 A g^−1^, in 3 M KOH electrolytes were used to test the cycling life stability of the N-rGO electrode.

The cycling stability test of the N-rGO electrode was accomplished by the GCD technique, and Equation (2) was used for performance calculations. The results exhibit consistent cycling life with almost 98.5% capacitance retention after 3000 charge/discharge cycles, as shown in Figure 9. During this cycling test, the active sites (nitrogen pyridine- and pyrrolic-like) may be lost, which could account for the slight decrease in specific capacitance value [27,60].

## 4. Conclusions

In this research, facile synthesis of nitrogen-doped reduced graphene oxide is introduced using a one-pot solvothermal process with the assistance of ethylene glycol (EG) as an organic solvent. In the reduction process, the solvent has a comprehensive effect on removing the oxygen-related functional group from GO networks. The performance of the synthesized N-rGOs was confirmed by XPS, XRD, FTIR, Raman, and FE-SEM analyses. The maximum concentrations of nitrogen and oxygen were 3.41% and 5.89%, respectively, after XPS analysis, suggesting that the sample was highly reduced and N-rich. The electrode prepared from N-rGOs displayed desirable charge accumulation characteristics with a high capacitance of 420 F g^−1^ at 1 A g^−1^ current density in 3 M KOH electrolyte and provided suitable cyclability. Furthermore, capacitance retention of 98.5% after 3000 charge/discharge cycles was achieved. The as-synthesized high-performance (N-rGOs) materials can be described as highly reduced, doped, and porous nanoparticles with a wide surface area, favorable carrier mobility, and sufficient wettability. Hence, this solvothermally prepared N-doped reduced graphene oxide has potential to be employed for future energy storage improvements.

## Figures and Tables

**Figure 1 nanomaterials-13-00666-f001:**
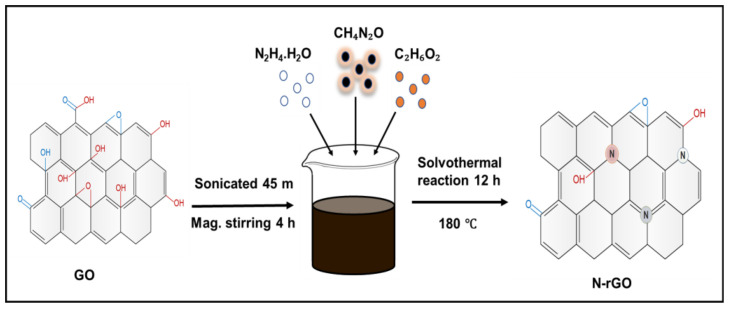
The illustrative layout depicts the procedure to prepare N-rGO nanoparticles.

**Figure 2 nanomaterials-13-00666-f002:**
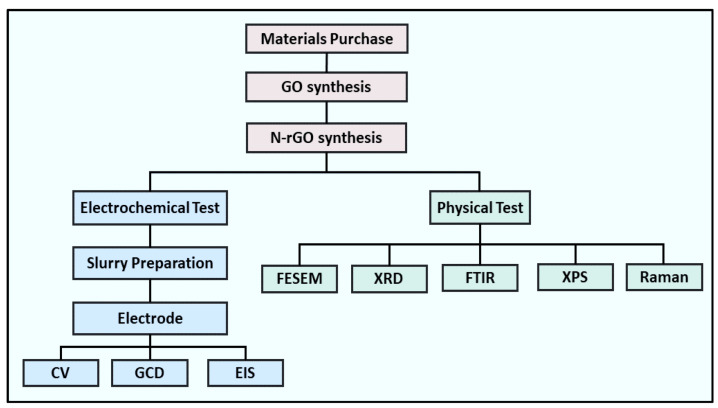
The flowchart displays the entire process of the optimization and performance tests of electrode materials.

**Figure 3 nanomaterials-13-00666-f003:**
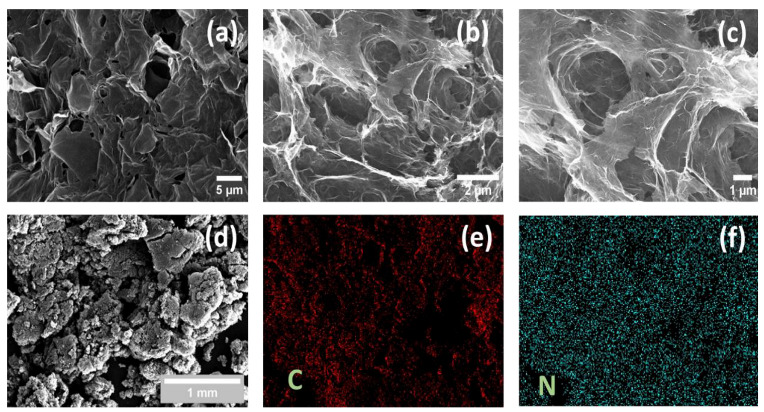
The images from FE-SEM analysis: (**a**,**d**) for GO; (**b**,**c**) for N-rGO-1 and N-rGO-2, respectively; the elemental mapping of (**e**) carbon and (**f**) nitrogen for N-rGOs.

**Figure 4 nanomaterials-13-00666-f004:**
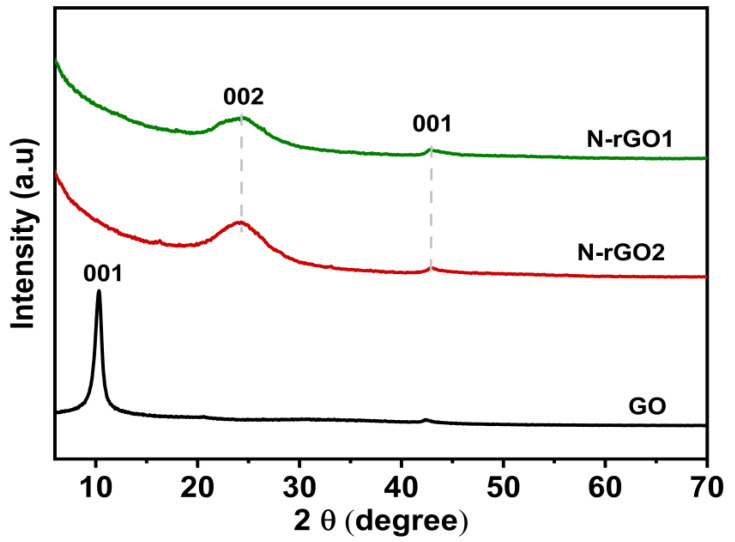
The graphical demonstration of the XRD plot of GO and N-rGOs.

**Figure 5 nanomaterials-13-00666-f005:**
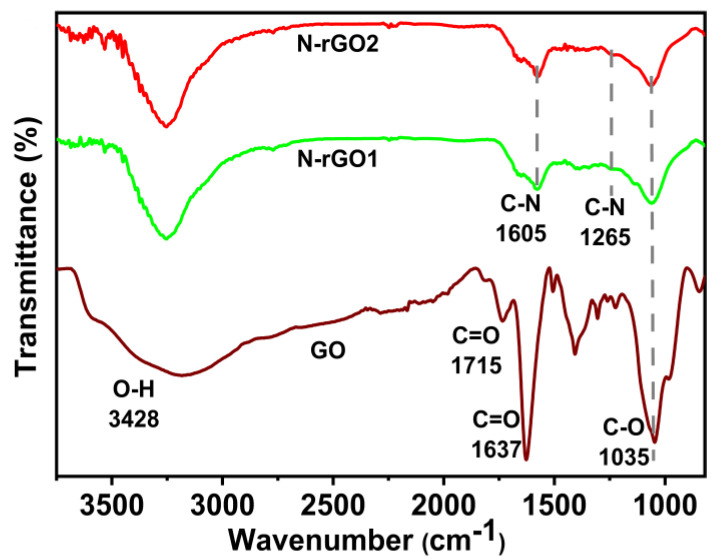
The graphical representation of the FTIR graph of GO, N-rGO-1, and N-rGO-2.

**Figure 6 nanomaterials-13-00666-f006:**
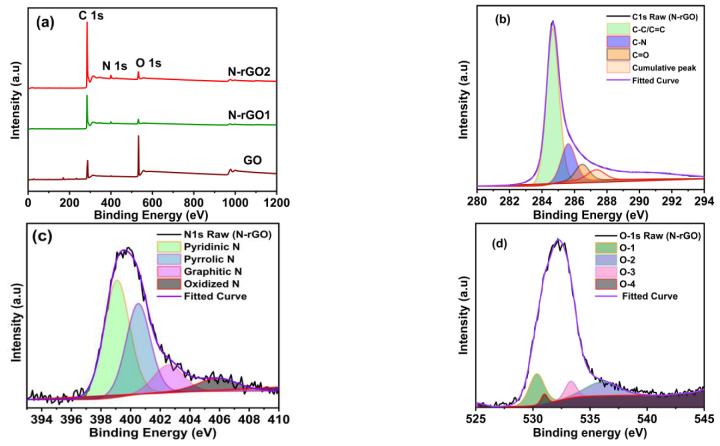
The XPS graph shows (**a**) the survey spectrum of GO and N-rGOs, (**b**) c1s, (**c**) N1s, and (**d**) O1s of N-rGOs.

**Figure 7 nanomaterials-13-00666-f007:**
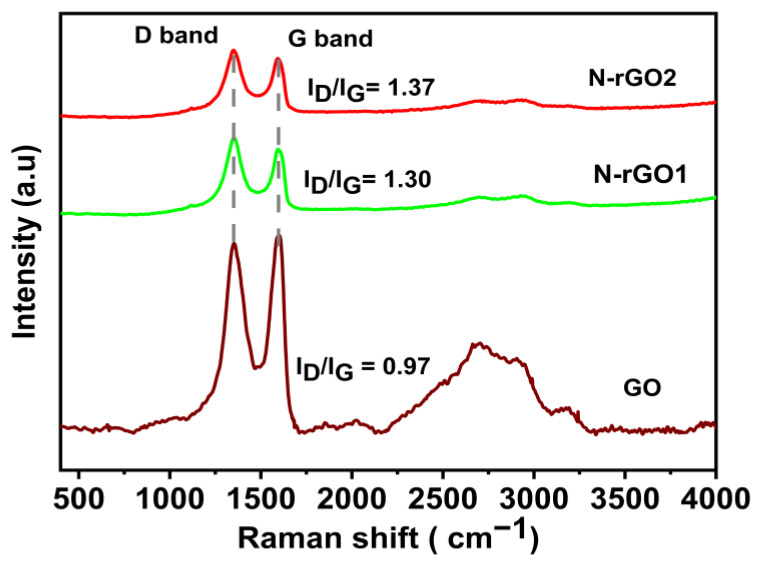
The Raman shift exhibited spectra of GO, N-rGO-1, and N-rGO-2.

**Figure 8 nanomaterials-13-00666-f008:**
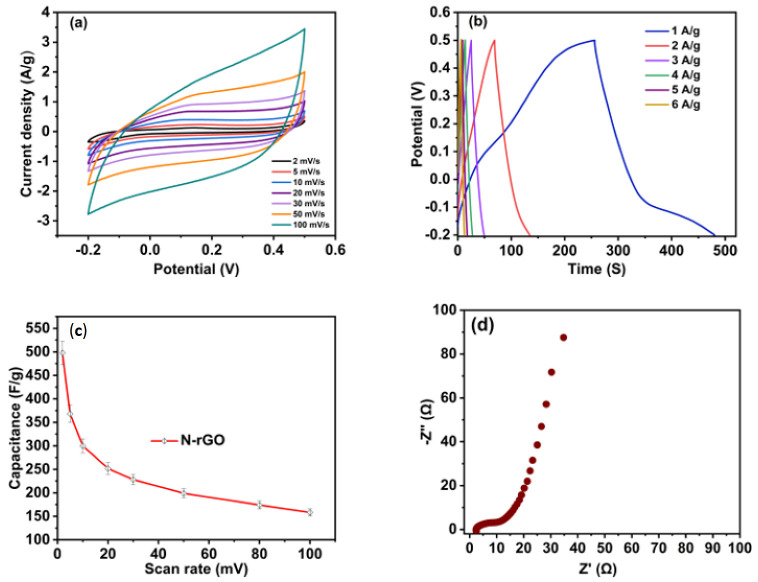
Electrochemical properties: (**a**) CV curves of N-rGOs, (**b**) GCD with several current densities, (**c**) variable capacitance with different scan rates, (**d**) EIS plot of N-rGO-based electrode.

**Figure 9 nanomaterials-13-00666-f009:**
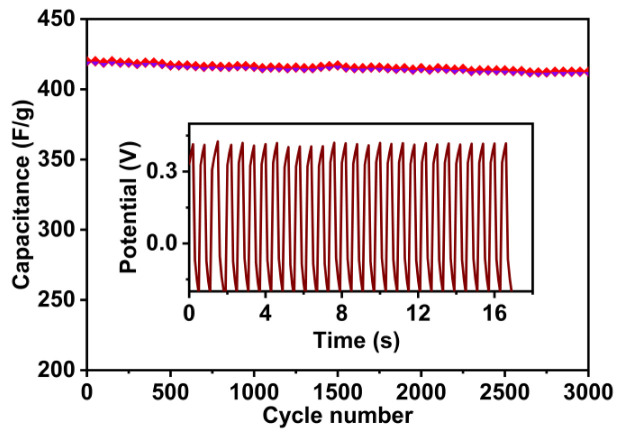
Capacity retention performance after 3000 charge/discharge cycles.

**Table 1 nanomaterials-13-00666-t001:** Presence of carbon, nitrogen, and oxygen contents in GO and N-rGO samples based on XPS analysis data.

Samples	C (%)	O (%)	N (%)
GO	64.74	35.26	-
N-rGO-1	90.70	5.89	3.41
N-rGO-2	90.27	6.42	3.31

**Table 2 nanomaterials-13-00666-t002:** Electrochemical performances of reduced graphene oxide-based electrodes made with various synthesis processes.

Material	Synthesis Process	Electrolyte	Current Density (A/g)	Specific Capacitance (F/g)	Ref.
NG	Solvothermal	6 M K.O.H.	0.1	301	[57]
N-rGO	Hydrothermal	BMIMBF_4_	1	390	[50]
NG sheet	Hydrothermal	6 M K.O.H.	5	295	[47]
NDG	Pyrolysis	1 M H_2_SO_4_	0.8	220.5	[58]
TRGO	Pyrolysis	EMIMBF_4_	1	290	[59]
NRGO	Hydrothermal	1 M H_2_SO_4_	0.1	199	[60]
NGA	Hydrothermal	1 M H_2_SO_4_	0.2	223	[61]
NGS-HMT	Hydrothermal	6 M K.O.H.	0.5	161	[22]
N-pGr	Hydrothermal	P.V.A.:H_2_SO_4_	1	230	[62]
rGO	Solvothermal	5 M KOH	0.25	183	[63]
Ti3C2	Solvothermal	3 M K.O.H.	2.5	119 F/cm^3^	[64]
S-GN	Electrochemical	3 M K.O.H.	3	320	[18]
N-rGO	Solvothermal (EG)	3 M K.O.H.	1	420	This work

## Data Availability

Not applicable.

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
