# Peer review of "One-Step Solvothermal Synthesis by Ethylene Glycol to Produce N-rGO for Supercapacitor Applications"

_nanomaterials, 2023, doi:10.3390/nano13040666_

Round 1
Author Response
Dear respected reviewer
Good day
Thank you very much for the issues and suggestions. we have made necessary changes in the manuscript. We think we fulfilled the requirements and that is well reflected in the revised version of the manuscript.

Reviewer 2 Report
In this manuscript, N-doped reduced graphene was synthesized by ethylene glycol (EG) assisted solvothermal method, and the N- doped reduced graphene was tested as electrode materials for supercapacitors. However, the use of EG as N source to produce N-doped graphene with solvothermal method is not new, and many reports had been published for application in supercapacitors. These kinds of reports can be found about 5 to 8 years ago, such as https://doi.org/10.1016/j.apsusc.2015.07.123, https://doi.org/10.1016/j.jpowsour.2014.12.025, https://doi.org/10.1007/s11581-017-1975-3 , https://doi.org/10.1007/s10008-018-4086-9 . There is no new science or phenomenon in this manuscript, so I suggest rejection.
Comment:
1. In this manuscript, the author calculated the energy density and power density. Actually, the energy density and power density can only be calculated by a two-electrode full cell. Only three-electrode test method was used in this manuscript, so it is not right.
Author Response
Dear reviewer
Thank you very much for your valuable comments. We have taken all the comments in our full consideration and have made necessary changes accordingly that is reflected in our revised version and we hope that will satisfy you.

Reviewer 3 Report
The manuscript entitled "One-step solvothermal Synthesis by ethylene glycol to produce N-rGO for supercapacitor applications" reported a novel electrode for supercapacitors using an efficient and facile approach to produce nitrogen-doped reduced graphene oxide (N-rGO) nanoparticles with an organic solvent ethylene glycol (E.G.) which effectively reduced epoxy groups in graphene networks through a solvothermal synthesis process. In addition, this one-pot facile solvothermal synthetic process would be an efficient route to develop electrode materials for future-generation supercapacitors. In general, it is an interesting and valuable topic to deserving a research article.
However, there are still many problems to be solved. So this reviewer would suggest a major revision before its acceptance.
1. Overall the draft is good but needs more careful editing.
2. Excel figures must not contain the superfluous outer frame. The same information must not be duplicated in Tables and Figures.
3. When generally introduce the supercapacitors, some important and recent review articles should be included: Nanocellulose and its derived composite electrodes toward supercapacitors: Fabrication, properties, and challenges; Design and fabrication of conductive polymer hydrogels and their applications in flexible supercapacitors; Emergence of melanin-inspired supercapacitors; Recent progress in carbon-based materials for supercapacitor electrodes: a review; etc.
4. Some of the content in Figure 3 is not clear (Figure 3.d.), and the reviewer would like the author to submit a clearer version in the new manuscript.
5. Error bars should be added to some of the results of multiple tests to conform to statistical science.
6. Graphene have been reported as additives for the electrode for its advantages, which should be further clarified in the manuscript with supporting articles: Energy Technology 8 (9), 2000397, 2020; Journal of Colloid and Interface Science 599, 443-452, 2021; etc.
7. In “3.6. Electrochemical performances”, the “3 M KOH” format is different from the other electrolyte formats mentioned earlier, and it is hoped that the author will compare it in the chart with other work using 3 M KOH electrolyte.
8. The Figures should be in the same format. Regroup the Figures and replace some parts of the Figures that are not obvious.
9. In Figure 9, the author mentions that 3000 charge-discharge cycles have been done, but the figure does not reflect the obvious reduction of capacitor retention rate. The 3000 cycles are too few. If the author has done this work, please submit it in the latest manuscript.
10. The relevant content of supercapacitor should be compared with the latest research, and should be represented in Figures or one table. Here the reviewer provides the author with some literatures and suggests the author to cite them in the latest version of the manuscript: Polymers 14 (13), 2521, 2022; New Journal of Chemistry 45 (48), 22602-22609, 2021; Polymer 235, 124276, 2021; Chinese Chemical Letters 31 (7), 1986-1990, 2020; Diamond and Related Materials 129, 109339, 2022; Diamond and Related Materials 128, 109283, 2022; Diamond and Related Materials 130, 109526, 2022; Journal of Colloid and Interface Science 609, 179-187, 2022; Diamond and Related Materials, Volume 131, January 2023, 109582; etc.
11. Please carefully check the whole manuscript. There are still some typos and grammar issues. In addition, please carefully check the references to ensure the full information is included.
Author Response
Dear respected reviewer
compliment of the day
thank you very much for your suggestions. we have taken all the issues in our full consideration and made necessary changes in the manuscript. The responses are uploaded for your kind perusal.

Reviewer 4 Report
This manuscript by Prof. Rahman et al. describes the nitrogen-doped reduced graphene oxide (N-rGO) nanoparticles with an organic solvent ethylene glycol (E.G.) through a solvothermal synthesis process as the N-rGO-based electrode.
There are some major issues to be addressed by the author before the acceptance by the Journal.
- In lines 175-716, the SEM images of two samples of N-rGO-1 and N-rGO-2 were shown. However, in Sec. 2.2 N- rGO synthesis needs detailed information for the preparations of N-rGO-1 and N-rGO-2.
- What is the function of N2H4.H2O for the N- rGO synthesis?
- In line 201, a tiny broad peak for contemplation (001) cantered at 2=42.42 was observed. The 2=42.4 should be a typo.
- Lines 350-351 were “assigned to the reduced restacking and porous nanoparticles with wide surface area, excellent carrier mobility, and good wettability.” Why?
There are no BET and contact angle measurements given in the manuscript.
Author Response
Dear reviewer
Thank you very much for your valuable comments. We have taken that in our full consideration and made necessary changes accordingly that is reflected in our revised version. We hope revision will make you satisfy.
Thank you

Round 2
Reviewer 2 Report
All the questions have been addressed. I suggest this paper can be accepted.